# Development and Characterization of Ammonia Removal Moving Bed Biofilms for Landfill Leachate Treatment

**DOI:** 10.3390/microorganisms12122404

**Published:** 2024-11-23

**Authors:** Rossana Petrilli, Attilio Fabbretti, Kathleen Pucci, Graziella Pagliaretta, Valerio Napolioni, Maurizio Falconi

**Affiliations:** 1School of Biosciences and Veterinary Medicine, University of Camerino, Via Gentile III Da Varano, 62032 Camerino, MC, Italy; rossana.petrilli@unicam.it (R.P.); attilio.fabbretti@unicam.it (A.F.); valerio.napolioni@unicam.it (V.N.); 2Eco Control Laboratorio Ascolano s.r.l., 63900 Fermo, FM, Italy; tecnico@ecoelpidiense.it; 3Eco Elpidiense s.r.l., 63821 Porto Sant’Elpidio, FM, Italy; biotecnologie@ecoelpidiense.it

**Keywords:** biological nitrogen removal, moving bed biofilm (MBB), landfill leachate, microbial community, metagenomics

## Abstract

Urbanization growth has intensified the challenge of managing and treating increasing amounts of municipal solid waste (MSW). Landfills are commonly utilized for MSW disposal because of their low construction and operation costs. However, this practice produces huge volumes of landfill leachate, a highly polluting liquid rich in ammoniacal nitrogen (NH_3_-N), organic compounds, and various heavy metals, making it difficult to treat in conventional municipal wastewater treatment plants (WWTPs). In recent years, research has shown that microbial biofilms, developed on carriers of different materials and called “moving bed biofilm reactors” (MBBRs), may offer promising solutions for bioremediation. This study explored the biofilm development and the nitrification process of moving bed biofilms (MBBs) obtained from high ammonia-selected microbial communities. Using crystal violet staining and confocal laser-scanning microscopy, we followed the biofilm formation stages correlating nitrogen removal to metagenomic analyses. Our results indicate that MBBs unveiled a 10-fold more enhanced nitrification rate than the dispersed microbial community present in the native sludge of the Porto Sant’Elpidio (Italy) WWTP. Four bacterial families, Chitinophagaceae, Comamonadaceae, Sphingomonadaceae, and Nitrosomonadaceae, accumulate in structured biofilms and significantly contribute to the high ammonium removal rate of 80% in 24 h as estimated in leachate-containing wastewaters.

## 1. Introduction

Urbanization and population growth have raised the issues of the production and, consequently, the need for appropriate treatment of increasing amounts of municipal solid waste (MSW). Because of low building and operation costs, landfills are widely used in Italy, as well as in developed countries, for MSW disposal. Inevitably, this generates huge volumes of landfill leachate, a highly pollutant liquid containing elevated concentrations of ammonium nitrogen (NH_4_^+^-N) and organic compounds in addition to a large spectrum of heavy metals [1,2,3,4,5]. Currently, most of the leachates from Italian landfills are classified as intermediate (5–10 years old) and mature (≥15 years old) leachates [3]. They are characterized by a low content of biodegradable organics due to two major factors: (a) their long stay in landfills; and (b) the introduction of separate waste collection, which occurred in Italy about a decade ago. Thus, these leachates show a low biochemical oxygen demand/chemical oxygen demand ratio (BOD/COD ≤ 0.3) and elevated concentrations of ammonia (2000–4000 mg/L). Leachates with such a chemical composition are very hard to treat in conventional municipal wastewater treatment plants (WWTPs) where nitrogen removal is mainly promoted by the microbial community of the activated sludge [2,6]. In fact, the nitrification-denitrification process involves first the conversion, by autotrophic ammonia-oxidizing bacteria (AOB), of ammonium ion (NH_4_^+^) to nitrite (NO_2_^−^), which is further oxidized to nitrate (NO_3_^−^) by nitrite-oxidizing bacteria (NOB). Finally, NO_3_^−^ is reduced to gaseous nitrogen compounds by anoxic/anaerobic bacteria, which use organic carbon as an energy source [7].

In the last two decades, many studies have explained why and how planktonic cells start colonizing living and/or inert surfaces and move to a biofilm lifestyle. A biofilm displays a three-dimensional structure with a highly ordered and complex extracellular matrix [8,9]. The development of a mature biofilm confers to the cell population better fitness and protection against a broad range of environmental stress conditions including antimicrobial agents and sanitation processes of food and water. Thus, biofilm formation can generally increase the biological risk to human health [10,11,12]. On the other hand, bacterial biofilms are currently regarded as very promising, innovative, and powerful tools in bioremediation techniques. In fact, microbial communities organized into biofilms are much more active than planktonic cells in degrading harmful pollutants such as inorganic and organic compounds, plastic, and metals. According to this, microbial proliferation and biomass development are induced on carriers of different materials, which are subsequently placed into the reactor. This technology, called a moving bed biofilm reactor (MBBR), was first invented in Norway [13] and is now employed worldwide for the treatment of industrial and municipal wastewater. Biofilm reactors can operate in aerobic, anoxic, or anaerobic conditions, and it has been extensively proven that microbes attached to carriers and organized in multi-layered biofilms show an enhanced nitrification-denitrification activity as compared with dispersed biomass. In fact, stratified biofilms allow, through an oxygen gradient, the simultaneous growth of nitrifiers and denitrifiers for complete nitrogen removal, whereas organic matter is degraded by aerobic bacteria located mostly in the outer layers of biomass [14,15,16]. Thus, MBBRs are very efficient in ammonia removal in pollutants containing a low carbon/nitrogen (C/N) ratio, such as those in landfills [17,18,19,20]. Unlike suspended activated sludge reactors, the moving bed biofilm process also permits the volumetric capacity of the tank through the concentration of biomass to be fully exploited, thereby maximizing the plant’s performance. In addition, MBBRs offer the advantages of reducing the clogging of bio-carriers and the number of head loss events because they do not require sludge recycling [21,22,23].

In a recent study, the bacterial population of the activated sludge derived from the WWTP of Porto Sant’Elpidio (FM, Italy) was subjected to repetitive ammonium stresses (NH_4_^+^-N ≅ 350 mg/L) in a minimal medium supplemented with leachate as only carbon source. These experiments, named the Repetitive Re-Inoculum Assay (RRIA) [24], allowed us to obtain an optimized microorganism mixture that can efficiently abate the high concentration of ammonia contained in mature landfill leachates. Herein, the formation of moving bed biofilms from these NH_4_^+^ selected microbial communities was investigated using crystal violet staining (CVS) and confocal laser-scanning microscopy (CLSM). Then, the nitrification process promoted by MBBs was estimated in a leachate-based medium and correlated to metagenomic analyses performed by 16S rRNA gene next-generation sequencing. Taken together, our results reveal that four bacterial families, Chitinophagaceae, Comamonadaceae, Sphingomonadaceae, and Nitrosomonadaceae, were predominant on structured biofilms derived from high NH_4_^+^ stress-selected bacterial consortia and largely contribute to the excellent ammonium removal rate (≅80% in 24 h) as estimated in wastewater–leachate mixtures.

## 2. Materials and Methods

### 2.1. Characteristics of Biofilm Carriers

The K1 moving bed biofilm carrier (Kaldnes, Sandefjord, Norway), used in Bio-RRIA experiments (see below), is characterized by polyethylene high density (HDPE) hollow cylinders with an internal cross and 18 external fins; the diameter is 9.1 mm, and the length is 7.2 mm. The density is 140 kg/m^3^, and the efficient surface area is 500 m^2^/m^3^.

Limited to LIVE/DEAD staining and confocal laser-scanning microscopy analysis, the carriers were HDPE sheets that were 15-micron thick and 1 cm^2^ in size.

### 2.2. Preparation of Bio-Carriers

Bacterial adhesion to K1 carriers and HDPE sheets was obtained using both selected bacteria populations (G1-2, G5-2, and G5-4) and native sludge. Aliquots of glycerol stock G1-2, G5-2, and G5-4 were plated on solid agar (1.7%) leachate minimal medium (LMM) (see below) at the final concentration of NH_4_^+^ of 350 mg/L. After 2 days of incubation at 30 °C, the bacterial lawns formed were collected with a sterile cotton swab and suspended in 30 mL of LMM. Then, 30 pieces of K1 carriers were immersed in the bacterial suspension. The initial optical density (OD_600_) was ≅0.3, corresponding to ≅5 × 10 ^7^cells/mL as determined by plating bacteria on Luria–Bertani (LB) solid medium after serial dilutions. The cultures were incubated in a rotary shaker at 28 °C for 10–15 days, and the decrease of ammonia (NH_3_) was monitored with Nessler’s test. Since ammonia degradation by bacteria caused acidification (pH about 6.5) of cultures, the initial pH = 8 of the LMM medium was restored by adding NaOH (5 N). The dissolved oxygen (DO) concentration was measured using the Edge DO meter (Hanna Instruments, Smithfield, RI, USA) and kept at 5.8 ± 0.5 mg/L with mild agitation. The composition, at the family level, of the selected bacterial populations G5-2 and G5-4 is reported in Appendix A. For sludge bio-carriers, 10 mL of the activated sludge (150 mg, dry weight) was mixed with 20 mL of the LMM and 30 pieces of K1 carriers and incubated at 28 °C following the above-described protocol. The optical density of this culture could not be measured for the high turbidity due to the addition of the sludge, and only the colony-forming units (CFUs) were determined (≅2 × 10^6^ cells/mL).

The LMM was prepared by combining 60 mL of leachate with 30 mL of MM medium and 90 mL of distilled water. This step causes a 3-fold dilution of raw leachate. The MM broth was composed of three solutions that were mixed together. Solution 1 contained 4.95 g of (NH_4_)_2_SO_4_, 0.62 g of KH_2_PO_4_, 0.27 g of MgSO_4_, 0.04 g of CaCl_2_, 0.5 mL of FeSO_4_ (30 mM in 50 mM EDTA at pH 7.0), 0.0002 g of CuSO_4_, and 1.2 L of distilled water. Solution 2 contained 8.2 g of KH_2_PO_4_, 0.7 g of NaH_2_PO_4_, and 3 L of distilled water, which was brought to pH 8.0. Solution 3 contained 0.6 g of Na_2_CO_3_ and 12 mL of distilled water. The three solutions were sterilized using filtering, and the leachate was sterilized using an autoclave. The ammonia concentration of bacterial cultures was adjusted to the values indicated in Figure Legends, adding 1 M of (NH_4_)_2_SO_4_. The average compositions of pre-treated leachates used in our experiments are reported in Appendix A.

For determination of colony-forming units (CFUs) from bio-carriers, the HPDE supports were subjected to pulsed low-intensity sonication (three cycles of 45 s at 3.5 Watts) in 3 mL of ice-cold 0.9% NaCl solution to completely remove the bacterial biofilm. Then, the cells, after adequate dilutions, were plated on LB solid medium and incubated overnight at 28 °C.

### 2.3. Crystal Violet Staining

Crystal violet staining was carried out essentially as previously described [25,26]. Briefly, K1 bio-carriers were rinsed with 0.9% NaCl solution to eliminate medium residual and incubated with 0.1% (*w*/*v*) crystal violet solution in distilled H_2_O. After 15 min at room temperature, the bio-carriers were rinsed four times with distilled H_2_O to remove unbound dye. The crystal violet was washed out from the biofilms with 4 mL of 95% ethanol, and the color intensity was quantified spectrophotometrically at a wavelength of 600 nm [27].

### 2.4. LIVE/DEAD Staining and Confocal Laser-Scanning Microscopy (CLSM)

The HDPE sheets were fluorescence stained using the LIVE/DEAD^®^BacLightTM Bacterial Viability Kit according to manufacturer protocol (Invitrogen, Carlsbad, CA, USA). Briefly, the two dyes, SYTO9 green fluorescent and propidium iodide (PI) red fluorescent, were applied simultaneously as a 1:1000 dilution in distilled water and incubated for 15 min at room temperature in a dark room. After washing the stained carriers with distilled H_2_O to remove the dyes in excess, the fluorescent signals from both live (green) and dead (red) bacteria were visualized simultaneously by a confocal laser-scanning microscopy CLSM (Nikon C2Si using the objective PLA APO λ 100× Oil, Nikon, Tokyo, Japan).

### 2.5. Analytical Methods

Concentration determinations of total nitrogen, ammonia, nitrite, nitrate, biochemical oxygen demand (BOD), and chemical oxygen demand (COD) were carried out as previously reported [24].

### 2.6. Analysis of Microbial Communities by 16S rRNA Gene Next-Generation Sequencing

Bacterial cells from native sludge (aliquots of 10–20 mL) were harvested by centrifugation (8000 rpm for 20 min), whereas bio-carriers (4–6 tubes) were pulled out of culture medium, washed in 0.9% NaCl solution and subjected to a mild sonication for cells detachment as described above. Then, chromosomal DNA was extracted from all samples by the E.Z.N.A. Kit (Omega Bio-tek Inc., Norcross, GA, USA) according to the instructions given by the manufacturer. DNA concentration was estimated by NanoDrop (ThermoFisher Scientific, Waltham, MA, USA), and the V3–V4 hypervariable regions of 16S rDNA were amplified by PCR using universal primers 341F 5′-TCGTCGGCAGCGTCAGATGTGTATAAGAGACAGCCTACGGGNGGCWGCAG-3′ and 805R 5′-GTCTCGTGGGCTCGGAGATGTGTATAAGAGACAGGACTACHVGGGTATCTAATCC-3′. Then, 16S rRNA gene next-generation sequencing by MiSeq Illumina Platform (San Diego, CA, USA) was performed as previously reported [24]. Operational taxonomic units (OTUs) were defined by clustering at a minimum pair-wise identity threshold of 97%. The NCBI 16S RefSeq database was employed for taxonomic classification.

## 3. Results and Discussion

### 3.1. Monitoring and Quantification of Biofilm Formation

In a recent study, financed by the Eco Elpidiense s.r.l., a private company that manages the municipal wastewater treatment plant of Porto Sant’Elpidio (FM, Italy), we developed a bacteria selection procedure named the Repetitive Re-Inoculum Assay (RRIA). This assay, somewhat resembling the dilution to extinction approach [28], consists of consecutive re-inocula/dilutions (7–10 times) of the same bacterial community in a liquid leachate-based medium (LMM) at elevated NH_4_^+^-N content (≅350 mg/L). RRIA caused a drastic reduction (from ≅250 to ≅15) of bacterial species present in the activated sludge, causing a remarkable enrichment of those microorganisms characterized by elevated tolerance to recurring ammonia stresses coupled with high activity in nitrogen removal. Aliquots of RRIA cultures were taken at different times during the 2-month assay for bacterial species identification (Appendix A) using metagenomic analysis and cell storage at −80 °C [24].

In the present work, we first established whether our selected microbial communities were able to develop organized biofilms. To this end, K1 HDPE bio-carriers were prepared from both native sludges and RRIA-derived multi-species suspensions to verify the microbial adhesion to these plastic supports by the crystal violet staining (CVS). This colorimetric method is easy, quick to perform, and cost-effective, although it has the disadvantages of being poorly quantitative and not distinguishing viable and non-viable cells [29]. The result, shown in Appendix A, was very promising and revealed that bacteria of all the processed samples stably colonized a large surface of K1 polyethylene supports, as visible from the intense blue color. In fact, G5-4, G1-2, and sludge bio-carriers, after ethanol wash and measure of related optical densities, displayed values of A_600_ nm about six times higher than that of controls (carriers not immersed in cell cultures), suggesting that this plastic material was a suitable carrier to be used in our planned MBBs experiments. In line with Irankhah et al. [30], who investigated MBB reactors from mixed cultures, we found comparable optical densities (A_600_ nm ≅ 2.0) in CVS assays in addition to the common observation of evident cell adhesion at the air–liquid interface of the walls of the flasks in all experiments.

Furthermore, biomass accumulation was also estimated as a function of time during biofilm formation, for G5-2 and native sludge HDPE carriers, using LIVE/DEAD staining and confocal laser-scanning microscopy (CLSM) (Figure 1). For image acquisitions, bio-carriers were taken out from bacterial suspensions in LMM broth on the 5th, the 10th, and the 20th days following the ammonia degradation. On the 20th day, the remaining bio-carriers were re-inoculated in a fresh medium (NH_4_^+^-N concentration ≅ 350 mg/L), and incubation was prolonged for an additional 10 days at 28 °C. The selective staining coupled with CLSM clearly visualized the predominance, across all samples, of green fluorescent–labeled cells (living ones), demonstrating that bacteria were abundant and, for a very large majority, vital; otherwise, red fluorescent cells (dead ones) were almost completely absent. According to the random acquisition of images, the cell number seemed to be slightly higher for G5-2 than in sludge MBBs in the first 20 days of biofilm development, whereas such difference disappeared for longer times (25th and 30th days) when more structured biofilms were formed. Colony-forming units (CFUs), determined by growing bacteria on LB agar plates after serial dilutions, supported the CLSM analysis and revealed that the cell number was similar for both biofilms, which was ≅ 4 × 10^3^ per HPDE sheet on the 5th day and progressively increased up to ≅5 × 10^4^ per HPDE sheet on the 30th day. Sludge and G5-2 MBBs also showed comparable partial nitrification activity. Thus, the NH_4_^+^-N concentration diminished from ≅350 to ≅60 mg/L in 6 days, corresponding to a removal rate of 83%, which was consistent with that estimated with K1 bio-carriers in the initial phase of the Bio-Repetitive Re-Inoculum Assay (see next paragraph)**.**

### 3.2. Ammonia Degradation by Biofilms Developed from Ammonia-Selected Bacterial Populations and Activated Sludges

Given that the biofilm-forming ability and biomass measurement were ascertained for our mixed cultures by different methods (Appendix A and Figure 1), G5-2 and G5-4 MBBs were tested for their capability to degrade ammonia in a minimal broth (LMM) supplemented with leachate as only carbon source (Figure 2). These bio-carriers were inoculated in LMM medium, and the ammonia concentration was estimated at the starting point (time 0) and at regular time intervals for the next 21 days. As bacterial biofilms began to oxidize ammonia and its concentration dropped to ≅80 mg/L (falls), bio-carriers were removed, washed, and transferred to a fresh leachate broth so that the initial value of NH_4_^+^-N (≅360 mg/L) was restored (peaks). This step was repeated 10 times. This experiment is essentially the Repetitive Re-Inoculum Assay (RRIA) as previously described by Petrilli et al. [24], with the exception that MBBs were used instead of planktonic cells as in the original assay. Thus, it was named Bio-RRIA. As seen in Figure 2A, the two different MBBs showed basically matching nitrogen removal patterns throughout the entire duration of the assay, characterized by an initial adaptation period in which a lower efficiency in ammonia degradation was observed. In fact, the NH_4_^+^ concentration was reduced from ≅360 to ≅80 mg/L (removal of 78%) in 5 days. From the 6th to the 15th days, the removal rate progressively increased and peaked, which identified the transfer of bio-carriers in a fresh leachate medium with consequent extra NH_3_ stress, became closer and closer. Thus, after five re-inocula (on the 15th day), MBBs degraded the same amount of ammonia (78%) in only 24 h, exhibiting a 10-fold higher removal activity than the respective original microbial cultures. In fact, when G5-2 and G5-4 bacterial populations were grown in the planktonic phase, under the same experimental conditions, about 9–11 days were necessary to obtain ≅75% ammonia oxidation [24].

During the last two decades, the performance of many biofilm-based reactors has been extensively investigated, and a bibliometric analysis reveals that ammonia removal varies approximately between 60 and 95% depending on the biofilm system adopted, the type of wastewater, ammonium strength, and bio-carried used. Most of these bioreactors operated with an initial ammonia concentration of less than 100 mg/L [18,33,34,35,36,37,38]. In this context, our results are promising, given that G5-2 and G5-4 MBBs show an excellent removal rate (≅80% in 24 h) with high-strength ammonium (≅360 mg/L). According to our standard protocol for the preparation of bio-carriers, the Bio-RRIA assays were highly reproducible with high ammonia-selected MBBs, and occasionally, small differences, limited to the length of the initial adaptation phase, were found. Another Bio-RRIA, in which the three biofilms, G5-2, G5-4, and G1-2, were compared, showed that, after the usual adaptation period, the one-day removal rate of 67% was achieved (Appendix A). Nitrification was slightly lower due to the higher concentration of ammonia (≅425 mg/L) used in this Bio-RRIA. 

In addition to NH_4_^+^, the levels of the most common nitrogenous compounds were estimated during the ammonia oxidation by the bacterial biofilms (Figure 2B,C). A progressive accumulation of NO_2_^−^ was observed for both G5-2 and G5-4 MBBs, whereas appreciable amounts of NO_3_^−^ (≥50 mg/L) were produced only in the final part of the Bio-RRIA (after the 15th day). Consistently, the total nitrogen (NT) was only slightly reduced as a function of time, and its level closely accounted for the sum of the ammonia remaining, nitrites, nitrates, and non-biodegradable organic compounds from leachate. Under the experimental conditions used, leachate-based medium and presence of oxygen (DO ≅ 5.8 ± 0.5 mg/L), these findings indicate that MBBs mainly promoted nitrification due to the predominant carrier adhesion, selective growth, and action of autotrophic and heterotrophic nitrifying bacteria (see metagenomic analysis). Importantly, biofilm enables bacteria with different nutritional requirements to coexist by occupying distinct layers and niches within stratified biofilms where inorganic/organic carbon, oxygen, and various substrates can be exchanged by bacteria. In this context, multicomponent biofilms, as utilized in our experiments, are more efficient than single-species biofilms in bioremediation. 

The municipal wastewater treatment plant (WWTP) of the Porto Sant’Elpidio town is located in central Italy near the Adriatic Sea coast where the climate is very mild and, particularly in recent years due to global warming, the temperature in winter rarely falls below 5–10 °C. A detailed description of this WWTP has been previously provided [24]. Thus, the effect of the temperature on the nitrification process by G1-2, G5-2, and G5-4 biofilms was investigated in the range of 10–30 °C (Figure 3). For this purpose, bio-carriers were taken from the final phase (≅after 20 days) of a preparative Bio-RRIA where biofilms were already formed and the ammonia removal rate was optimal (≅78% in 24 h). Then, MMBs were placed in a fresh leachate-based medium to start a new Bio-RRIA at four different temperatures. The nitrogen removal rate remained constant at 30 °C and 23 °C but slowed down by ≅ three times (78% in 72 h) at 18 °C for all the samples. In contrast, biofilm nitrification activity was considerably reduced at 10 °C (75% in 14 days). Importantly, the wastewater temperature in the two nitrification/denitrification tanks (1450 m^3^ each) of the Porto Sant’Elpidio WWTP drops down 15–20 °C only in January and February, as monitored by the Eco Elpidiense (FM, Italy), the private company that manages this municipal wastewater bioreactor. Thus, the finding that G1-2, G5-2, and G5-4 MBBs retain their maximum performance at 23 °C with a limited loss of activity at 18 °C is very encouraging for their immediate application in the pilot reactor of 2000 L, which is already available at the Eco Elpidiense Company. Depending on the pilot reactor results, we are confident that our approach derived from MBB technology, implemented and transferred on a large scale, could be used to improve, in the near future, the Porto Sant’Elpidio municipal bioreactor performance by accelerating the biological conversion of ammonia to nitrite. This reaction is surely the most critical step of the entire nitrification-denitrification process by bacteria [39] and, importantly, needs a lot of energy to make the aeration system, which injects air into the tanks of the WWTP, operate. 

Colorimetric assays and CFUs (Figure 1 and Appendix A) showed that, in addition to high ammonia-selected cultures, also the bacterial community present in the native sludge from the Porto Sant’Elpidio MWWTP was able to adhere and form biofilm on HDPE supports. To obtain a representative picture, native sludges were sampled during the last 2 years, considering the chemical analyses performed by the Eco Control Laboratory (FM, Italy), to estimate the municipal bioreactor performance. In fact, the treatment capacity of a municipal plant can vary due to many factors (i.e., season, rainfall, change of equivalent people, etc.). However, in the specific case of the Porto Sant’Elpidio bioreactor, the treatment capacity also depends on the quantity and quality of landfill leachate introduced (GP and KP, personal communication). As compared with the high reproducibility of G5-2 and G5-4 MBBs, Bio-RRIA carried out with different sludge-derived biofilms exhibited quite variable ammonia degradation curves (Figure 4 and Appendix A). In fact, the LS1 MBB plot was characterized by an initial ammonia removal rate of ≅60% in 10 days, a value more than two-fold lower than that of G5-2 and G5-4 MBBs. After this acclimatization period, a progressive increase of the NH_4_^+^ oxidation activity and consequent shortening in time was observed for G5-2 and G5-4 biofilms (Figure 2A). Diversely, SL4 MBB displayed a very poor nitrification capacity (NH_4_^+^ removal of ≅50% in 15 days) that remained low for the entire length of Bio-RRIA. Accordingly, this sludge sample was withdrawn when the Porto Sant’Elpidio WWTP was malfunctioning, and dense and persistent foams were well visible on the surface of wastewater contained in the two nitrification/denitrification tanks. As seen in Figure 4B,C, better NH_4_^+^ degradation patterns were obtained with SL2 and SL3 sludge-derived biofilms that approached the elevated ammonium removal rate of G5-2 and G5-4 MBBs. In fact, SL2 and SL3 biofilms lacked a pronounced early adaptation phase and were characterized by an average removal of ≅76% in 5–6 days that did not change for the duration of the Bio-RRIA experiment (30–40 days). Altogether, these results suggest that, unlike the high ammonia-selected G1-2, G5-2 and G5-4 MBBs, native sludge biofilms display a large variability and are much less reliable in producing biofilms with high efficiency in ammonia oxidation. As seen in Appendix A, some sludge MBBs could be partially functional or even unsuccessful.

### 3.3. Composition of Biofilms Formed from Ammonia-Selected Microbial Communities and Activated Sludges

The microbial composition of G5-2 and G5-4 biofilms, used in the Bio-RRIA experiments, was investigated as a function of time using 16S rRNA NGS analysis. According to the near-coincident NH_4_^+^ degradation curves (Figure 2A), the metagenomic analysis of these MBBs led to comparable results in terms of bacterial species identified, except for slight differences in their relative abundance. As seen in Figure 5A,B, at the family level, Chitinophagaceae, Comamonadaceae, Sphingomonadaceae, and Nitrosomonadaceae substantially increased over time, becoming the most prevalent in both G5-2 and G5-4 biofilms. In particular, Chitinophagaceae, which did not exceed 1.5% in liquid bacterial cultures and newly formed MBBs (the 1st day), reached 13% and 9% for G5-2 and 25% and 15% for G5-4 on the 10th and 15th day, respectively. Similarly, the Nitrosomonadaceae fraction, which was less than 1% in the first stages of biofilm development (on the 1st day), became 16–18% in both MBBs after 10 days. Comamonadaceae and Sphingomonadaceae changed to a lower extent in G5-2 and G5-4 samples, showing a two- to five-fold increment in structured biofilms (on the 10th and 15th days) as compared with planktonic growth cells and MMBs on the 1st day. By contrast, Alcaligenaceae and Xanthomonadaceae families, which were strongly predominant (≅10–30% of all OTUs) in the early points of Bio-RRIA, dramatically decreased in MBBs for longer times (abundance ≤ 5% on the 10th and the 15th days). Notably, except for Chitinophagaceae, the bacterial families identified in our study were also found by Garcia et al. [40], who investigated in bench-scale MBB bioreactors how size and geometry of plastic supports affect biofilm structure, function, and relative microbial species abundance. Most of these families include many heterotrophic nitrifying bacteria, and Alcaligenaceae and Comamonadaceae, particularly, have been shown to perform simultaneous nitrification/denitrification in MBB reactors [41]. The biofilm composition at the genus level of G5-2 and G5-4 samples is reported in Appendix A. 

Consistent with the taxa distribution at family and genus levels, we identified three principal bacterial species that exhibited a huge increase in frequency at the more advanced stages of biofilm formation and were reasonably responsible for the progressive increment of the nitrification process (Figure 5C,D). Primarily, *Nitrosomonas eutropha*, one of the most investigated AOBs, initially at the limit of detection, reached 18% of all OTUs in both G5-2 and G5-4 MBBs on the 15th day. *Nitrosomonas* sp. was found to tolerate elevated NH_4_^+^ concentrations [42], and numerous studies indicated that this bacterium plays a key role in NH_4_^+^ degradation in different types of biofilm reactors for wastewater treatment [43,44,45,46,47]. Remarkably, *N. eutropha* is an autotrophic species that oxidizes ammonia in the presence of oxygen. In this respect, MBBs utilized in our Bio-RRIAs were operating in aerobiosis but in heterotrophic conditions due to the huge quantity of organic matter contained in leachate. Thus, the successful proliferation of *N. eutropha* can be explained by the fact that organized biofilms constitute a physicochemical barrier which, by allowing a selective penetration of nutrients (i.e., oxygen, inorganic carbon, and nitrogen compounds), creates an optimized micro-environment for growth and activity of this autotrophic nitrifier. Furthermore, a considerable biomass accumulation was observed on the 10th also for two genera of Chitinophagaceae. In fact, *Ferruginibacter lapsinanis* came up to 6% and 11% on G5-2 and G5-4, respectively, and *Flavitalea flava* to 8% on G5-4. Chitinophagaceae includes many heterotrophic AOB that oxidize ammonia using as an energy source the organic matter contained in leachate, and it was one of the most represented families identified by Wang et al. [48] studied for several months the nitrogen removal in a pilot-scale reactor. Altogether, our results suggest that, as discussed above, G5-2 and G5-4 biofilms, unlike planktonic cells, can provide multilevel metabolic environments that permit concomitant autotrophic/heterotrophic nitrification. Finally, *Comamonas faecalis*, although identified at all times, reached 12% on the biofilms after 10–15 days, possibly leading to the conversion of nitrate, first to nitrite, and then, although at a low extent, to gaseous nitrogen compounds. In fact, the denitrification activity of bacteria belonging to this genus has been known for a long time [49], and recently, it has been demonstrated in removing several pollutants that produce NO_2_^−^ and NO_3_^−^ as by-products of their degradation [50,51,52,53]. Unexpectedly, the *Nitrospira* and *Nitrobacter* genera, including canonical nitrite-oxidizing bacteria and generally prevalent in activated sludges from MWWTPs [54,55,56], were not identified across all MBBs analyzed in this study. The lack of these NOBs is possibly due to the unfavorable conditions (high concentrations of ammonium and organic compounds) used in biofilm formation and to the competition with heterotrophic species. Finally, other clades, such as *Castellaniella hirudinis*, *Paracoccus koreensis*, *Paracoccus pantotrophus*, and *Stenotrophomonas daejeonensis*, were considerably represented at the starting points, while became almost undetectable for longer times of Bio-RRIA. In particular, the drastic OTUs fall of *C. hirudinis* and *S*. *daejeonensis* accounted for the disappearance, at the family level, of Alcaligenaceae and Xanthomonadaceae, respectively (Figure 5).

It is well established that, in most cases, the poor performance of municipal bioreactors is caused by significant alterations of the microbial population of activated sludges in terms of bacterial species and their relative abundances. Thus, we correlated the ammonia degradation efficiency of SL2, SL3, and SL4 biofilms derived from the Porto Sant’Elpidio MWWTP sludge with their taxa distribution as determined using the metagenomic analysis. As evident in Figure 6, at the family level, SL3 MBB was quite dissimilar from the other two sludge biofilms. Specifically, SL3 MBB, which was very effective in NH_4_^+^ removal (Figure 4), exhibited the uncommon prevalence (≅16% of all the OTUs) of Zoogloeacea and the almost complete lack (<1%) of Nitrosomonadaceae. Zoogloeacea was not detected either in sludge SL2 and SL4 MBBs or in G5-2 and G5-4 MBBs formed from NH_4_^+^-selected microbial communities. Nevertheless, Zoogloeacea accounted for 2.6% of total bacterial communities of activated sludges, as shown by metagenomic analyses carried out in 14 MWWTPs in Asia and North America [57]. Notably, the two genera *Zoogloea* and *Thauera* represented the totality of members of this family in SL3 biofilm, being 12% and 4%, respectively, as estimated by metagenomic analysis. *Zoogloea* sp. is a heterotrophic/aerobic denitrifier that performs nitrogen removal when cultured in the presence of nitrates and nitrites. In addition, *Zoogloea*. sp. was also found to exhibit a considerable nitrification capacity (NH_4_^+^ removal rate of 44%) without nitrate and nitrite production when it was grown in a high ammonia-based medium [58]. Thus, the predominance of *Zoogloea* can account for the elevated ammonia oxidation potential of SL3 MBB in the absence of *Nitrosomonas* sp. The other two families, the Chitinophagaceae and Comamonadaceae, which accumulated in G5-2 and G5-4 biofilms, were identified at considerable levels, as well as in sludge MBBs. In particular, Chitinophagaceae, constituted by the only two genera *Ferruginibacter* and *Flavitalea*, became the most abundant family in SL2 (≅10%) and SL4 (≅16%) MBBs, whereas Comamonadaceae was equally represented (≅4–6%) across all sludge biofilms (Figure 6). The possible role of these taxa in bioremediation has already been discussed relative to G5-2 and G5-4 MBBs. As described in Section 3.2, the SL4 biofilm had a reduced efficiency in ammonia degradation (Figure 4D) according to the fact that sludge sampling was done when the Porto Sant’Elpidio WWTP was not well-performing, and an abundant foam was produced during the wastewater treatment. Notably, SL4 MBB contained considerable levels of two genera, *Gordonia* (4.5%) and *Rubinisphaera* (4.7%), which were not identified in the other sludge biofilms. While very little is known about *Rubinisphaera* (Planctomycetaceae), several members of *Gordonia* (Gordoniaceae) display a powerful catabolic activity in degrading hazardous pollutants such as organic substances, hydrocarbons, sulfur compounds, and rubber. The selection of this genus might have been caused by the peculiar and transient conditions related to the malfunctioning of the Porto Sant’Elpidio WWTP at the sampling time. Currently, *Gordonia* is emerging as a promising candidate, particularly in microbial consortia, to be used in targeted bioremediation techniques [59]. According to the SL4 sample, *Gordonia* strains were detected in stable foams formed in wastewater treatment plants [60,61]. Foams denote the suffering of the bioreactor and usually create, as observed for the Porto Sant’Elpidio WWTP, serious operating problems. In addition, SL4 MBB showed a lower content than SL2 MBB (6% vs. 9%) of Nitrosomonadaceae, which could be another reason that somewhat explains the malfunctioning of this sludge biofilm in ammonia oxidation. These findings provide further evidence that the bacterial composition of sludge biofilms strongly affects the nitrification process. Thus, monitoring the sludge microbial community can help to predict with a certain advance and eventually act to limit the possible malfunctioning of the bioreactor.

## 4. Conclusions

The present study represents an advancement of our recent work [24] in which, through multiple ammonia stresses (Repetitive Re-Inoculum Assay), we strongly enriched the bacterial community of activated sludges to produce an optimized mixture of microorganisms retaining high efficiency in nitrogen removal from landfill leachate. Herein, we proved, using two different techniques (CVS and CLSM), that these high ammonia-selected microbial populations developed stable and organized biofilms on plastic carriers. Importantly, these bio-carriers (G1-2, G5-2, and G5-4 MBBs), after an initial adaptation phase, achieved a very good ammonia removal rate (≅80% in 24 h) in leachate-based medium with high-strength ammonium (≅360 mg/L). Metagenomic analysis revealed that four families, Chitinophagaceae (13–25% represented by the two genera *Ferruginibacter* and *Flavitalea*), Nitrosomonadaceae (16–18%, genus *Nitrosomonas*), Comamonadaceae (16–18%, genus *Comamonas*) and Sphingomonadaceae (5–7%, genus *Sphingopyxis*) accumulated in biofilms at advanced stages of development suggesting their crucial role in the nitrification process. Basically, MBBs exhibited a 10-fold higher activity in NH_4_^+^ degradation than respective planktonic growth bacteria due to the biofilm advantage of enabling a synchronous action of autotrophic and heterotrophic nitrifying bacteria.

Therefore, the performance of the Porto Sant’Elpidio MWWTP was monitored for more than 2 years, and MBBs were produced from the native sludge of this bioreactor. These biofilms evidenced huge variations in activity and reduced NH_4_^+^ degradation as compared with G1-2, G5-2, and G5-4 MBBs. Intriguingly, the most active sludge biofilm obtained (SL3) was characterized by an ammonia removal rate of 78% in 3 days and by a quite rare predominance (16%) of the Zoogloeaceae family. These results, particularly those achieved with the ammonia selection-derived MBBs, are very encouraging, and we are planning to quickly transfer this MBB-based technology initially to the pilot reactor (2000 L) available at the Eco Elpidiense Company and, subsequently, to the Porto Sant’Elpidio plant.

## Figures and Tables

**Figure 1 microorganisms-12-02404-f001:**
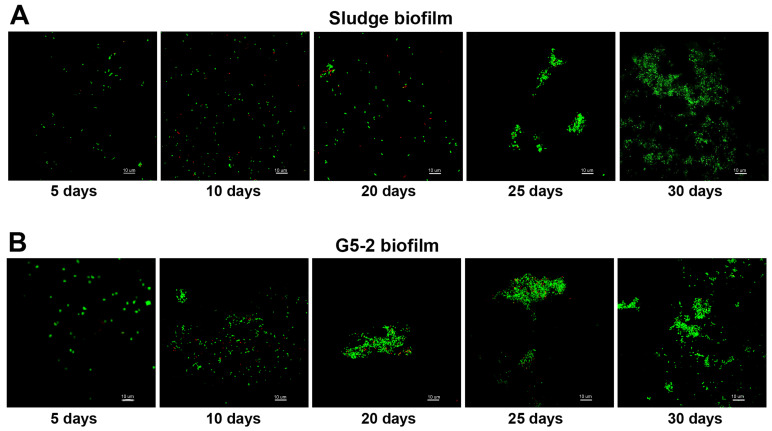
Analysis of biofilm formation using CLSM. Staining of sludge (**A**) and G5-2 (**B**) MBBs was carried out with SYTO9 and PI fluorescent dyes, followed by CLSM analysis as described in Section 2. Both dyes intercalate with nucleic acids but, while SYTO9 penetrates both living and dead cells, PI can only pass through damaged membranes and displaces SYTO9, allowing differentiation between live (green) and dead (red) cells [31,32]. Stacks of images were taken at random areas on the HPDE sheets, and each stack contained 10 images.

**Figure 2 microorganisms-12-02404-f002:**
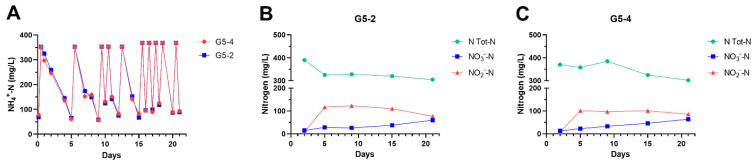
Bio-repetitive re-inoculum assay using G5-2 and G5-4 MBBs. Bio-carriers of G5-2 and G5-4 (30 pieces each) were incubated in 30 mL of LMM at 28 °C. The NH_4_^+^-N was adjusted to ≅350 mg/L and monitored, using Nessler’s test, as a function of time for the whole duration of Bio-RRIA (**A**) (see body text). Samples from Bio-RRIA were withdrawn at indicated NH_4_^+^-N falls for total nitrogen, nitrite, and nitrate determinations of G5-2 (**B**) and G5-4 (**C**) biofilms. Total nitrogen is the sum of ammonia remaining, nitrites, nitrates, and organic nitrogen compounds from leachate. Leachate composition is reported in Appendix A.

**Figure 3 microorganisms-12-02404-f003:**
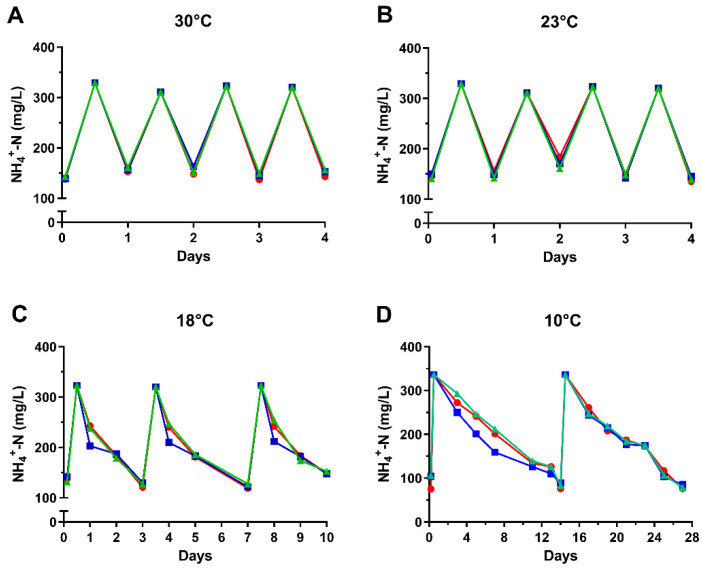
Effect of the temperature on Bio-RRIA by G1-2, G5-2 and G5-4 MBBs. The G1-2 (red circle), G5-2 (blue square), and G5-4 (green triangle) MBBs were incubated at 30 °C (**A**), 23 °C (**B**), 18 °C (**C**) and 10 °C (**D**). and then subjected to Bio-RRIA essentially as described in the body text and in the legend of Figure 2A.

**Figure 4 microorganisms-12-02404-f004:**
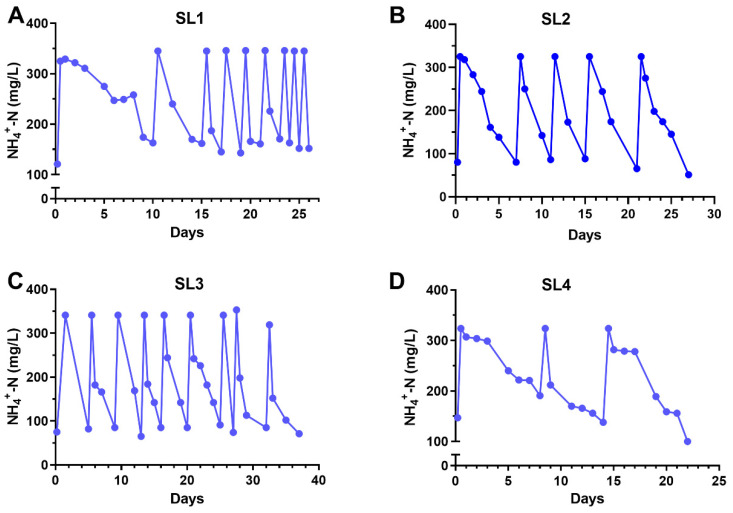
Bio-repetitive re-inoculum assay by sludge MBBs. Sludge samples were taken at different periods from the Porto Sant’Elpidio MWWTP and used for K1 bio-carriers preparation. Biofilms were investigated for their efficiency in NH_4_^+^-N removal in Bio-RRIA experiments performed as described in the body text and in the Legend of Figure 2A. Four representative sludges, indicated with SL1 (**A**), SL2 (**B**), SL3 (**C**) and SL4 (**D**), are shown.

**Figure 5 microorganisms-12-02404-f005:**
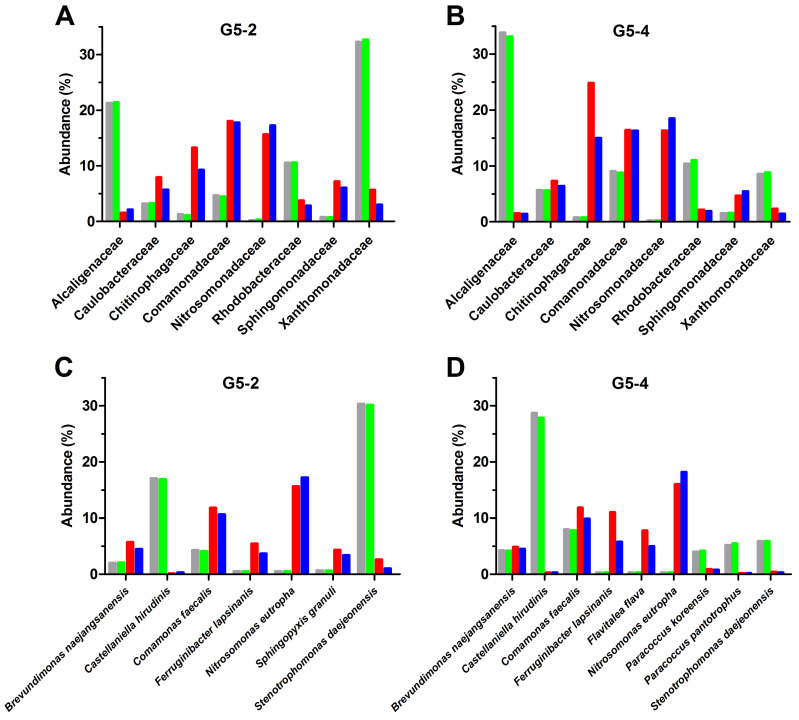
Change of biofilm population from selected microbial communities. Relative abundance, expressed as a percentage of total operational taxonomic units (OTUs), of the most prevalent families (**A**,**B**) and species (**C**,**D**) in the biofilms derived from G5-2 and G5-4 cultures are shown. Samples for species identification by 16S rRNA NGS were the bacterial cultures used for biofilm formation (gray bars) and MBBs taken on the 1st day (green bars), on the 10th day (red bars), and on the 15th day (dark blue bars) of Bio-RRIA reported in Figure 2A. The bar graph shows only families and species that contributed more than 4% to the total bacterial community in at least one point.

**Figure 6 microorganisms-12-02404-f006:**
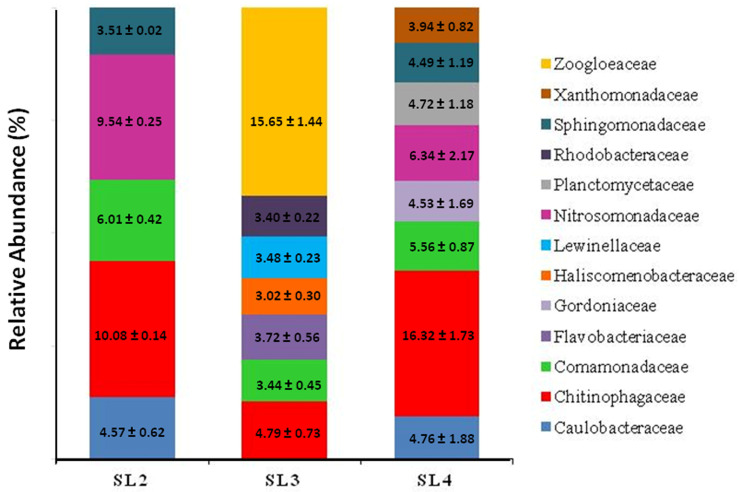
Stacked bar chart of MBBs from native sludges. Relative abundance, expressed as a percentage of total operational taxonomic units (OTUs), of the most prevalent families in biofilm-derived sludges SL2, SL3, and LSL4. Bacteria identification was carried out using 16S rRNA NGS, and values represent the average ± standard deviation of at least three points taken during Bio-RRIA experiments, as shown in Figure 4. Only families that contributed more than 3% to the total bacterial community are reported. Analysis of variance across the three samples shows a statistically significant difference in their microbial composition (*p* < 0.05).

## Data Availability

The original contributions presented in the study are included in the article and Appendix A, further inquiries can be directed to the corresponding author.

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
