# Peer review of "Development and Characterization of Ammonia Removal Moving Bed Biofilms for Landfill Leachate Treatment"

_microorganisms, 2024, doi:10.3390/microorganisms12122404_

Round 1

Reviewer 1 Report

Comments and Suggestions for Authors

The manuscript entitled “Development and characterization of ammonia removal moving bed biofilms for landfill leachate treatment” explored the biofilm development and the nitrification process of moving bed biofilms (MBBs) obtained from high ammonia-selected microbial communities. The results are interesting. However, I suggested that this work should be improved before publication. The specific comments are as follows:

From lines 25 to 26, I think the author would better provide microbial community structure on the genus level. A similar style was observed in the following parts, including the conclusion, figure 6.

In line 163, the author should add information on the method. For example, what primers does the author use? Did the author carry out PCR?

In line 164, I think the OTUs analysis is outdated. I suggested the author reanalyze the author's ASVs method.

In 174 and 175, why do the authors use “@”? A similar writing style was observed in the following parts.

Does the ultimate concentration meet the discharge standard? As I can see, the treated ammonium concentration is still high.

Does the author set repeat groups for the experiment? Because I did not see the error bars in the figures. The result also lacks statistical difference analysis. The results of statistical difference analysis are robust evidence to support your conclusion and are necessary in articles.

Comments on the Quality of English Language

no comments

Author Response

We thank the Reviewers and Editorial Board for the positive evaluation of our research and for their valuable suggestions to improve the manuscript. We hope that the revised version of the manuscript is now suitable for publication in Microorganisms.

All changes in the manuscript are indicated as follows:

Deleted text = struck through words

New text, corrections  =  red words

Reviewer 1

The manuscript entitled “Development and characterization of ammonia removal moving bed biofilms for landfill leachate treatment” explored the biofilm development and the nitrification process of moving bed biofilms (MBBs) obtained from high ammonia-selected microbial communities. The results are interesting. However, I suggested that this work should be improved before publication. The specific comments are as follows:

 From lines 25 to 26, I think the author would better provide microbial community structure on the genus level. A similar style was observed in the following parts, including the conclusion, figure 6.

Answer. Thanks for pointing this out. We decided to provide only the family level of the microbial community in the “Abstract” because of the highly experimental nature of our study. At this stage, we believe it would increase the chance of replicability of future studies in this field. However, the predominant genera (Ferruginibacter, Flavitalea, Comamonas, Sphingopyxis and Nitrosomonas) in biofilms derived from selected cultures (G5-2, G5-4) have been now indicated in the “Conclusions” of the revised version and they were already being reported in Figure 5 and Supplementary Figure S4. For the same reasons, we compared only at family level the three biofilm-derived sludges (LS2, LS3 and LS4) shown in Figure 6. These biofilms were quite different from each other and a lot of genera have been identified. Thus, since the Figure 6 would result very complicated and poorly informative, we chose to indicate and discuss the most important genera for each biofilm in the text at pags. 12 and 13.

In line 163, the author should add information on the method. For example, what primers does the author use? Did the author carry out PCR?

Answer. We already stated that the methodology applied in the 16S rRNA gene next-generation sequencing by MiSeq Illumina Platform was performed as previously reported (ref. 24). The required information has been added at pag. 4 of the revised version.

 In line 164, I think the OTUs analysis is outdated. I suggested the author reanalyze the author's ASVs method.

Answer. The introduction of ASV methods was marked by a debate about their utility. Although OTUs do not provide such precise and accurate measurements of sequence variation, they are still an acceptable and valuable approach. In one research study, Glassman and Martiny (https://pmc.ncbi.nlm.nih.gov/articles/PMC6052340/) confirmed the suitability of OTUs for investigating broad-scale ecological diversity. They concluded that OTUs and ASVs provided similar results, with ASVs enabling a slightly stronger detection of fungal and bacterial diversity. And their work indicated that even though species diversification can be measured more accurately with ASVs, the use of OTUs in well-constructed studies is generally valid to demonstrate diversification at broad scales.

Moreover, stating that OTUs analysis is outdated seems quite arbitrary. If you would check on PubMed the term “operational taxonomic units”, you would get on the left part of the webpage the histogram with the number of documents per year indexing this term…you will notice that in 2020 the term reached its peak (875 documents), while decreasing in 2023 to 603 documents. If you will search “amplicon sequence variants”, the term reached its peak in 2022 (515 documents), while decreasing in 2023 to 513 documents. Thus, even if we should acknowledge the limitation of OTUs analysis compared to ASVs, we cannot really deem the OTUs analysis outdated. Indeed, we think that the OTUs analysis is still valid, at least depending on the study design and the expected outcome.

We deemed OTUs analysis a good approach to determine the microbial composition of the samples analyzed.

 In 174 and 175, why do the authors use “@”? A similar writing style was observed in the following parts.

Answer. The problem could be due to the version of the word which you are using, please countercheck also in the PDF file. Instead of the symbol “@” we meant “APPROXIMATELY EQUAL TO”.

 Does the ultimate concentration meet the discharge standard? As I can see, the treated ammonium concentration is still high.

Answer. The question raised by the Reviewer is very important. In fact, Italian environmental regulations establish that the limit for ammonia wastewater content discharged into sewage network is ≤ 30 mg/L. Because of very high ammonia concentration (1500 mg/L), leachate cannot be directly introduced into a municipal wastewater treatment plant (WWTP). In fact, peaks of ammonium in the WWTP (100-150 mg/L) are toxic for bacteria of the activated sludge. After an appropriate dilution, our moving bed biofilms could be used to further reduce ammonia from 400 to 80 mg/L or lower in a pre-treatment bioreactor. At this point, the removal of ammonia in leachate can be continued by conventional activated sludge systems. This represents the ultimate goal of our study.

 Does the author set repeat groups for the experiment? Because I did not see the error bars in the figures. The result also lacks statistical difference analysis. The results of statistical difference analysis are robust evidence to support your conclusion and are necessary in articles.

Answer. We thank the Reviewer for this remark. For most of our experiments, we considered a biological triplicate, that is considering the three samples G1-2, G5-2 and G5-4 as reported in Figure 3 and Supplementary Figure S2 (only G5-2 and G5-4 in Figure 2). We also used three/four biological samples when testing the differences across the three different biofilm-derived sludges reported in Figure 6. In this case, we now report the Standard Deviation and we tested the difference in microbial composition using ANOVA.

Reviewer 2 Report

Comments and Suggestions for Authors

The stages of biofilm formation that correlate nitrogen removal with metagenomic analysis are presented in the reviewed manuscript. In general, the manuscript is well prepared, but I think some aspects should be described in more detail.

In the introduction, the authors write about landfill leachate as a significant problem. Did the concentrations of ammonium nitrogen in the working solutions correspond to those observed in the landfill leachate? Why then was the LMM prepared by combining 60 ml of leachate with 30 ml of MM medium and 90 ml of distilled water? To what extent was the ammonia concentration reduced by the dilution?

Part of the methodology should be extended. The information about the selected populations should be included directly in the manuscript and not as a reference to the literature.

Can more information be given about the procedure of the experiment itself?

Were the pH and oxygen concentration analyzed?

Author Response

We thank the Reviewers and Editorial Board for the positive evaluation of our research and for their valuable suggestions to improve the manuscript. We hope that the revised version of the manuscript is now suitable for publication in Microorganisms.

All changes in the manuscript are indicated as follows:

Deleted text = struck through words

New text, corrections  =  red words

Reviewer 2

The stages of biofilm formation that correlate nitrogen removal with metagenomic analysis are presented in the reviewed manuscript. In general, the manuscript is well prepared, but I think some aspects should be described in more detail.

In the introduction, the authors write about landfill leachate as a significant problem. Did the concentrations of ammonium nitrogen in the working solutions correspond to those observed in the landfill leachate? Why then was the LMM prepared by combining 60 ml of leachate with 30 ml of MM medium and 90 ml of distilled water? To what extent was the ammonia concentration reduced by the dilution?

Answer. The average concentration of ammonia in leachate, we used in our experiments is 1400 mg/L but decreases to about 1000 mg/L after autoclave sterilization. Crude landfill leachate can reach up to 4000 mg/L of NH3. We previously showed that bacteria of the native sludge retain good ammonia degradation up to 400 mg/L of NH3 and higher concentrations of ammonia are toxic for bacteria population of the sludge (ref. 24). For these reasons, leachate was diluted about 3-fold and Bio-RRIA experiments have been carried out in a range of 350-400 mg/L ammonia.

Part of the methodology should be extended. The information about the selected populations should be included directly in the manuscript and not as a reference to the literature. Can more information be given about the procedure of the experiment itself?

Answer. We thank the Reviewer for this important suggestion. In the revised version, the Table S1, reporting the bacterial composition of selected populations, has been added in Supplementary Material. Therefore, additional information about 16S rRNA gene next-generation sequencing has been provided.

Were the pH and oxygen concentration analyzed?

Answer. The pH of LMM medium was 8. When ammonia was removed by bacteria the cultures acidified (pH = 6.0) and the initial pH was restored by adding a solution of NaOH (5 N). Notably, when the ammonia removal rate increased in Bio-RRIA (please see, Fig. 2A after the 15th day), bio-carriers were transferred in a fresh LMM every 24 hours and no extra controls were necessary. Dissolved oxygen (O2) concentration was 5.8 ± 0.5 mg/L.

This information is now provided at pag. 3

Round 2

Reviewer 1 Report

Comments and Suggestions for Authors

no comments

Reviewer 2 Report

Comments and Suggestions for Authors

Manuscript can be published in present form.